



# Increasing Aerosol Optical Depth Spatial And Temporal Availability By Merging Datasets from Geostationary And Sun-Synchronous Satellites

Pawan Gupta[1], Robert C. Levy[1], Shana Mattoo[1,2], Lorraine Remer[3], Zhaohui Zhang[1,4], Virginia R. Sawyer[1,2], Jennifer Wei[1], Sally Zhao[5], Min Oo[6], V Praju Kiliyanpilakkil[1,2] Xiaohua Pan[1,4]

[1]NASA Goddard Space Flight Center, Greenbelt, MD, USA
[2]SCIENCE SYSTEMS AND APPLICATIONS INC
[3]Univesity of Maryland Baltimore County, Baltimore, MD, USA
[4]ADNET SYSTEMS INC
[5]University of Maryland, College Park, MD, USA
[6]Space Sciences and Engineering Center, University of Wisconsin, Madison, WI, USA

*Correspondence to*: Pawan Gupta (pawan.gupta@nasa.gov)

**Abstract.** This comprehensive study analyzed aerosol observations from six Low Earth Orbit (LEO) and Geostationary Earth Orbit (GEO) sensors. LEO sensors like MODIS and VIIRS, providing 1-2 daily global measurements, were contrasted with GEO sensors (AHI, ABIs), offering high-frequency data (~10 minutes) over specific regions. The Dark Target aerosol retrieval algorithm was applied to six sensors (3 LEO and 3 GEO), and their Level 2 aerosol optical depth (AOD) data were grided and merged into a quarter-degree latitude-longitude grid with a 30-minute temporal resolution. Validation of AOD at 550 nm against AERONET measurements across global locations showcased the merged product's robustness, revealing a global mean bias of approximately ±0.05, and 65.5% of retrievals fell within an expected uncertainty range with a correlation coefficient of 0.83, underlining the reliability of the dataset. The new grided level 3 dataset significantly improved daily global coverage to nearly 45%, overcoming the limitations of individual sensors, which typically range from 12% to 25%. Furthermore, the study emphasized the unprecedented ability of the merged dataset to approximate the diurnal cycle of AERONET AODs, offering insights into unexpected diurnal signatures. The resulting dataset's high spatiotemporal resolution and improved global coverage, especially in regions covered by GEO sensors (Americas and Asia), make it a valuable tool for diverse applications. Tracking aerosol transport from phenomena like wildfires and dust storms gains precision, enabling enhanced air quality forecasting and hindcasting. Additionally, the study positions the merged dataset as a significant asset for evaluating and inter-comparing regional or global model simulations, previously unattainable in such a gridded format. The dataset and fusion framework layout in this study has the potential to include data from recently (future) launched other GEO (FCI, AMI) and LEO (PACE, VIIRS-JPSS) sensors.

## 1   Introduction and Motivation

Operational satellite remote sensing of aerosol properties and the dissemination of standard aerosol products is entering its third decade. Beginning with the launch of Terra in December 1999, the research and applications communities have been able to access a robust daily representation of Earth's global aerosol system at a variety of georeferenced spatial resolutions or on a 1 degree global grid. In particular, the Dark Target (DT) aerosol algorithm (Remer et al., 2005, Levy et al. 2013) has been applied to observations from the MODerate resolution Imaging Spectroradiometer (MODIS) on the Terra and Aqua satellites (Kaufman et





al. 2002) and later from the VIsible InfraRed Suite (VIIRS) on board the Suomi-National Polar orbiting Partnership (S-NPP) satellite (Sawyer et al., 2020) and now NOAA-20 (formerly known as JPSS-1). This algorithm introduced the research and applications communities to consistent, validated daily aerosol information. The information is provided as spectral aerosol optical depth (AOD) over dark, cloud-free, non-snowy, non-glint ocean and land surfaces, and for ocean scenes only, there is also

information on aerosol particle size (Levy et al., 2013). The DT standard products have become essential inputs to assimilation systems (Zhang et al., 2008, Benedetti et al., 2009, Gelaro et al., 2017), used for climate model validation (Kinne et al., 2003, Bellouin et al., 2008, Chin et al., 2014), estimates of intercontinental transport of particles (Kaufman et al., 2005a, Yu et al., 2012, 2019), and have provided new insight into aerosol-cloud processes (Kaufman et al., 2005a, Koren et al., 2007, 2008, Yuan et al., 2011ab). The AOD products are used as a proxy for particulate matter (PM), an essential air quality parameter (Al-Saadi et al.,

2005, Gupta and Christopher, 2009ab, Xin et al., 2014). Thus, the daily DT AOD aids operational air quality forecasting and post-event analysis of exceptional poor air quality events.

While the DT aerosol products cover the globe every day over mid and high latitudes and ever two days over the tropics, clouds, glint, and bright surfaces prevent the data sets from being truly global coverage. Analysis shows that over North America and

15 adjoining oceans, only about a third of all available scenes (~10 km x 10 km) produce an AOD product from a MODIS-like instrument ~~with 1 x 1 km spatial resolution~~ on any given day (Remer et al., 2012). Having multiple polar-orbiting views of the same scene might increase data product availability, but not much if the two instruments pass close in time, such as do Terra, Aqua, and S-NPP in North America. However, a sensor in geostationary orbit (GEO) makes many observations of the same scenes during the day, which gives more opportunities to observe that scene in cloud free/glint free conditions and return an AOD value at least

20 once per day. In fact, we see that geostationary sampling can increase the possibility of aerosol retrieval sometime during the day to nearly 100% for scenes that are cloudy at the MODIS overpass but are otherwise retrievable (Remer et al., 2012). If the interest in AOD products is simply to have one good retrieval per day, as offered by polar-orbiting satellites, then aerosol products from geostationary satellites will be sufficient to increase product availability and meet user needs.

Using one retrieval, taken at any time of the day, to represent daily conditions assumes that there is no significant diurnal signal at that site. On a global scale, this may be true. However, aerosols are dependent on source emissions, cloud processing, and local weather patterns, all of which have diurnal signatures. Previous studies suggest that aerosol diurnal signatures exist prominently at local scale, less so on regional scales (Kaufman et al., 2000, 2005c, Smirnov et al. 2002, Zhang et al., 2012, Arola et al. 2013 ). Therefore observing diurnal patterns are of interest to many research and applications communities. For example, the air quality

community can use for improvements to forecasting and mitigating unhealthy PM levels. The climate and cloud communities are interested in aerosol diurnal patterns because of the convolution between how aerosol and clouds evolve together during the day and how aerosol influences cloud and precipitation processes. These previous studies that found no regional diurnal aerosol patterns used ground-based remote sensing observations from the Aerosol Robotics NETwork (AERONET) (Holben et al., 1998, Giles et al., 2019). AERONET observations are site specific but globally distributed and report AOD every 5-15 minutes throughout the

daylight hours. Even though hundreds of AERONET stations went into the studies, the network still sparsely covers the globe compared to a satellite data set. One goal of producing aerosol products from geostationary satellite sensors will be the opportunity to determine possible aerosol diurnal signatures at a range of spatial scales, globally.



One caveat to using geostationary satellite observations for aerosol data sets is the loss of the global picture. By definition, geostationary satellites are regional instruments hovering over a specific equator position and imaging a circular portion of the Earth that covers roughly 1/3 of the global surface area. Covering the globe with multiple geostationary sensors is not the same as viewing the entire globe using multiple orbits of a single MODIS-like sensor. Not all geostationary sensors are identical, as

5     described in Section 3. Even if the sensors are built identically, they may not produce identical aerosol results, as was seen with Terra-MODIS and Aqua-MODIS (Levy et al., 2018). Thus, the ideal strategy to realize the benefits of geostationary aerosol data, namely increased availability and temporal resolution, without losing the benefits of daily global sampling of polar orbiting sensors, is to combine the aerosol products from both types of instruments. To do so, the strategy must be to maintain consistency in aerosol retrieval as it is applied to multiple sensors while accommodating the unique characteristics of each sensor.

In this paper, we report on a unique global data set produced from merging aerosol products from three polar-orbiting satellites (Terra-MODIS, Aqua-MODIS, S-NPP-VIIRS) and three geostationary satellites (Advanced Himawari Imager (AHI) on Himawari-8, Advanced Baseline Imager (ABI) on GOES-16, ABI on GOES-17). All six aerosol databases were produced with adapted versions of the DT aerosol algorithm, thereby maintaining consistency in the aerosol algorithm. The result is a global

gridded data set at 0.25 degree latitude/longitude spatial resolution and 30 minute temporal resolution that increases overall AOD availability with some ability to discern the diurnal cycle. Note that this is the first time that the DT aerosol product from multiple sensors has been gridded to a finer resolution than the MODIS 1 degree Level 3 operational product. In Section 2, we describe the DT algorithm and how it is adapted to different sensors and then describe how these individual data sets are gridded and integrated into a global grid. Section 3 demonstrates the increased availability of the aerosol product and expanded global coverage. This is

followed by comparing retrievals from the geostationary sensors to those from the polar-orbiting sensors, illustrating global consistency in the product. The new gridded data is validated compared with AERONET, and then diurnal signatures are explored regionally and globally. We wrap up the paper with a discussion of the utility of the new product, a discussion of aerosol diurnal signatures, and the outlook moving forward. Throughout the paper we will refer to geostationary instruments as GEO satellites or sensors and polar-orbiting instruments that fly on Low Earth Orbit satellites as LEO satellites or sensors.

## 2    Dark Target Aerosol Retrieval Algorithm

The DT algorithm has been documented extensively in the refereed literature (Kaufman et al., 1997, Tanré et al. 1997, Remer et al. 2005, Levy et al., 2013, Sawyer et al. 2020) and with a detailed description maintained on-line as an Algorithm Theoretical

Basis Document (ATBD, 2023. Here we provide a summary of the algorithm and then discuss the need for adjustments to the basic algorithm because of specific characteristics of new sensors.

The basic DT aerosol algorithm uses a Look Up Table (LUT) structure to link measured top of atmosphere (TOA) reflectance to an AOD value, given a specific sun-sensor geometry and assumptions about the aerosol properties and surface beneath. There are

separate algorithm structures and assumptions for retrievals over the ocean versus those over land. Over ocean, the algorithm uses up to six measured reflectances from wavelengths that span the range from 0.5 to 2.3 μm. The ocean surface is modeled as a rough ocean surface with ancillary data determining the surface wind speed and a constant spectrally-varying value used for water-leaving reflectance. The algorithm avoids clouds, sunglint, marine sediments and other ocean areas that do not conform to expected spectral signatures of the ocean surface. The aerosol properties over the ocean are modeled as individual lognormal modes of spherical





particles, some in the fine mode range and some in the coarse mode range. The algorithm finds the best combination of fine and coarse modes, and the number of particles in each one that produces the TOA reflectances best matching the corresponding spectral reflectances measured by satellite for the selected scene. The result is a solution set of spectral AOD that represents the aerosol loading, the modes chosen and the relative weight of each mode.

Over land the algorithm must address the complexity of a much more complicated surface than over ocean. The algorithm reduces uncertainties in the retrieval by choosing to retrieve only over "dark surfaces", mainly dark vegetated surfaces. Thus, the DT algorithm avoids clouds, snow, ice and non-vegetated deserts and other arid landscapes. Over the selected dark targets the DT land algorithm parameterizes the relationships between different wavelengths of surface reflectance using empirical constructs

10 (Levy et al. 2007). There is a special empirical relationship used for urban surfaces (Gupta et al. 2016). Over land, aerosol properties are modeled by bimodal representations. One of the bimodal models is coarse mode dominated, while the other three are fine mode dominated but with different absorption properties. The algorithm assigns specific absorption properties by region and season and then allows the retrieval to mix the assigned fine mode/specific absorption with the generic coarse mode model. Again, the aerosol loading is adjusted to match the measured TOA reflectances to results from the LUT. The primary result is the AOD at 0.55 μm.

The results of the DT algorithm, over land and ocean, are denoted as Level 2 products. The algorithm was initially developed for Terra and then applied to Aqua with no changes, as MODIS-Terra and MODIS-Aqua were designed to be identical twins. At first they produced identical results (Remer et al. 2006), but after some time sensor calibration began to drift and DT aerosol results began to diverge (Levy et al. 2018). However, when the DT algorithm was ported to VIIRS, its first non-MODIS application,

changes had to be made to the algorithm a priori (Sawyer et al., 2020). VIIRS uses different wavelengths than MODIS, even if the two sensors cover the same range, and VIIRS and MODIS have different swath widths and native pixel resolutions. Thus, the DT algorithm required calculation of a specific LUT for VIIRS to match the new wavelengths and required other modifications because of the change in native pixel resolution (Sawyer et al., 2020). In this study we port the DT algorithm again to new sensors (ABI and AHI), which require similar adjustments to the basic DT algorithm described here. The details of the instrument specifics

requiring algorithm adjustment are presented in Section 3.

### 3    Data and Methods

#### 3.1 Satellite Aerosol Datasets

The three LEO sensor data used in this study are two MODIS, one on Terra and one on Aqua and one VIIRS (on S-NPP). The three GEO sensors used are two ABIs (on GOES-16 and GOES-17) and AHI (on Himawari-08). For the period of this study (April 1, 2019, to March 31, 2020), GOES-16 was located in the operational GOES-East position (-75.2°W), GOES-17 in the GOES-West position (-137.0°), and Himawari-08 in the Himawari position (140.7°E). From each of these sensors, we use the Level 2

(L2) AOD retrieved by the DT algorithm, and refer to the products as MODIS-T, MODIS-A, VIIRS-SNPP, ABI-G16, ABI-G17 and AHI-H08, respectively.

For MODIS, we used DT retrieved AOD data at nominal 10x10 km$^2$ spatial resolution, specifically the standard MOD04/MYD04 collection 6.1 data set available via the Level-1 and Atmosphere Archive & Distribution System Distributed Active Archive Center


(LAADS DAAC). VIIRS AOD data used are retrieved at nominal 6x6 km$^2$ and we used the V1.1 (also available at LAADS). Note that since October 2022, there is new version V2.0 of VIIRS AOD retrievals, but they are not used here. The AOD data from all three GEO sensors (Gupta et al., 2019) are at nominal 10x10 km$^2$ and produced as version V0. From here on, we use the term "pixel" to refer to the size of a L2 retrieval box, noting that it varies between sensors, and varies across the scan or swath of each

sensor. A "native pixel" refers to the size of the original sensor observations (e.g. the Level 1 or L1 data).

Although the DT algorithm reports AODs in multiple wavelengths depending on individual sensors' available channels (table 2), we focus on AOD at 0.55 µm in this analysis. Quality flags (QF=0, 1, 2, 3) are produced with each retrieval, where 0 is marginal, 1 is good, 2 is moderate, and 3 is for best quality (highest confidence) data. We used each sensor's variable named

"Optical_Depth_Land_And_Ocean", which represents the Science Team's recommended method for data filtering of QA=2,3 over land and QA=1,2,3 over the ocean. Specifically, the following (Table 1) scientific data sets (SDS) from the Level 2 aerosol products from all six sensors are used in this study:

Table 2 provides details on each sensor, including channels, resolution, data version, and file specifications used in the current

analysis. It is important to note that each sensor stores the level 2 aerosol datasets in files covering different time windows. MODIS's L2 DT aerosols datasets come in 5-minute files (known as granules), VIIRS is in 6-minute granules, whereas all three GEO are in 10-minute files that represent Full-Disk (FD) observations. The coverage of data reported in each file from the six sensors also varies. Figure 1 demonstrates an example of spatial coverage from the individual sensors on September 7, 2019. The map is produced by plotting every fifth pixel from MODIS, every 10[th] from VIIRS and every third from GEO for better visuals of

coverage from an individual sensor. In this figure, coverage by the GEO sensor for 10-minutes is shown, where one single orbit (about 90-100 minutes) from the LEO sensor is shown. The two ABIs cover the western Hemisphere with some overlap, such that ABI-G16 has more coverage over the Atlantic Ocean, whereas ABI-G17 extends its coverage west into to the Pacific Ocean. The AHI full disk mostly covers Asia Pacific with some overlap with ABI west in the Pacific Ocean. The central time of observations from various sensors corresponds to 1200 UTC, which conveniently has Aqua and SNPP filling in over Africa. A few hours earlier

or later, the LEO sensors would overlap with the GEO, and Africa would have no observations.

### 3.2 AERONET Data

The Aerosol Robotic Network (AERONET) is NASA's global ground network of CIMEL sun-photometers that measure directly

transmitted solar light during daylight hours (Holben et al., 1998, Giles et al., 2019). The direct sun spectral measurements are used to derive aerosol optical depth at various wavelengths (340-1020 nm). The typical temporal resolution of AERONET is about every 15 minutes but varies with sun angles. Here we used Version 3.0, Level 2.0 (cloud screened and quality assured) AOD data. The Angstrom exponent is used to interpolate AERONET AODs at 500 nm and 675 nm to match with the satellite AODs at 550 nm. The uncertainty in AERONET AOD is of the order of 0.01-0.02. There are 387 global AERONET stations collocated with

satellite retrievals. The number of AERONET stations varies with individual satellites/sensors due to regional vs. global coverage from LEO and GEO sensors.

### 3.3 Level 2 Data Gridding and Integration



The primary goal of this study is to integrate the highest quality AODs retrieved using the DT algorithm from six sensors on a unified high-resolution grid by using every 30 minutes of observations. The output is a global gridded aerosol dataset with spatiotemporal resolutions of a quarter degree (0.25º x 0.25º) and 30 minutes.

The individual sensors are first gridded using the method developed for MODIS high resolution gridded datasets (Gupta et al., 2020). In this method, we start with L2 satellite data files (MYD04, MOD04, etc., table 2) and first group them into desired temporal resolution window (i.e., 30 minutes in this study; HH:00:00 – HH:29:59 or HH:30:00 - HH:59:59). In this way, the number of files that falls in the 30-minute window will vary from 3 for GEO sensors, 5 for VIIRS, and 6 for MODISs. The resulting 30-minute files will also have different spatial coverage for LEO sensors, but the coverage will remain fixed for GEO sensors. It

is important to note that the GEO sensor's aerosol dataset coverage will vary even though the field of view remains fixed because portions of the full disk will be in night and that dark portion advances diurnally. Similarly, the LEO sensors' aerosol retrieval coverage may also vary, due to whether observing the daytime or nighttime node of their orbits.

For a given sensor aggregated over the 30 minutes, we compute statistics for each of the Level 2 variables denoted in Table "1" on

our global 0.25°x0.25° grid (or quarter degree grid or QDG). The grids are identified by the latitude and longitude of the center of the grid cell. The gridding uses a box averaging method (Gupta et al., 2020) which accounts for the approximate size of the retrieval pixel (see Caviet 1- below). The result, for each variable, includes minimum, maximum, arithmetic average, standard deviation, and the number of level 2 pixels for each QDG. In this way, we store gridded AOD and angle statistics at each time stamp for any of the six sensors that have data in the 0.25º x 0.25º grid. The process is applied across the global grid.  Thus a grid square at a

particular time stamp may hold competing AOD statistics from 0, 1, 2, 3 or in highly unlikely events, 4 different sensors.

For a given time stamp and grid square, AODs from available sensors are then used to derive a merged AOD. Merge AOD statistics includes the simple mean (representing 1 or more sensors) the standard deviation, and number of sensors available for the QDG. There are no weighting functions (see Caveat 2). Thus, the final output file from this gridding process contains gridded AOD

statistics from the AODs of individual sensors plus merged AOD statistics calculated from the individual sensors for 30-minute time windows, globally. This way, we generate 48 files per day containing QDG global AOD data sets. Note that some variables (e.g. solar zenith angle) are relatively constant throughout the 30 minutes, and are sensor independent.

Caveat 1: A caveat of this simple gridding technique is related to the varying retrieval box sizes (pixels) of the different L2 product

from the different sensors.  Figure 2 shows examples of change in pixel area as a function of satellite viewing zenith angles for MODIS (top), VIIRS (middle), and ABIs, AHI (bottom). The pixel area is calculated while considering Earth's sphericity and for data presented in Figure 1. The actual change in the pixel area from nadir to the edge of the swath will vary as a function of latitude. As demonstrated in the figure, the pixel growth rate for GEO sensors (ABI and AHI) is much higher (by a factor of 2-20) than for LEO sensors (by a factor of 2-5). It is important to note that VIIRS has onboard pixel aggregation (Gladkova et al., 2016), which

does not allow its pixel to grow as much as MODIS pixels grow at higher viewing angles. MODIS and all GEO sensors do not have onboard aggregation to correct for bow-tie distortion (Sayer et al., 2015). Therefore, we are dealing with varying pixel sizes for each sensor with the added complexity of bow-tie distortion affecting each sensor differently. Due to this reason, a simple box averaging gridding process at a relatively higher spatial resolution can create artificial data gaps in the gridded data product as compared to level 2 swath data. Therefore, we use a spatial filling method (Gupta et al., 2020) by calculating pixel size as a function





of viewing angle and filling all the empty grids within the satellite's original pixel with the same value. This additional step brings the spatial coverage of gridded data to that of level 2 datasets.

Caveat 2: A second caveat is that in this implementation, there is no optimized weighting – in that there is no preference for using gridded AOD retrieved from one sensor over another. Whether there is one sensor, two, three, or more observing within a 0.25° grid or a 30 m time window, all individual sensor AODs are given the same weighting in the merge. A future version of this aggregation may include weights, with the weighting function based on validation (compared to AERONET), expected error based on uncertainty analysis, and/or another to-be-determined function.

## 4    Results and Discussion

The gridded AODs from six sensors plus the merged AODs are processed and stored in 30-minute global files following the method described in section 2. Figure 3 demonstrates an example of spatial coverage by each sensor and merged AOD for 30-minute gridded AOD data for March 26, 2020, at 2345 UTC (23:00 – 23:59 UTC). The top row shows three AOD maps for three GEO sensors, whereas the bottom row is for three LEO sensors. The light gray color on the map shows nighttime (SZA>90) during the time of observation. The GEO sensors provide full-disk coverage every 10 minutes, thus, the 30-minute AOD maps show an average of the three 10-minute full disk AOD retrievals. The circular data gap over the ocean is due to sun glint restriction (40° from specular angle) on AOD retrievals. The 30-minute coverage from LEO sensors (Figure 3, bottom row) covers only about 1/3 part of a satellite orbit, and the vertical data gaps in the middle of the satellite swath (orbit) are due to sun glint restrictions. The middle panel shows the merged AODs from all the sensors. The merged AOD maps clearly show that the AOD retrieved from six sensors is smoothly fused, and qualitatively there is no visible discontinuity in the spatial pattern from one sensor to another. The big circular holes in individual GEO sensors from sunglint are partially filled with another GEO sensor or an LEO sensor. Similarly, the glint shields in individual LEO sensors are also partially filled in by other sensors. In this example, none of the individual sensors provide complete AOD coverage of the daylight portion of the earth, but the fused dataset attempts to achieve that target limited by cloud/snow cover. The central map clearly demonstrates the primary goal of this dataset of providing aerosol observations every 30 minutes with nearly complete coverage of the daylight portion of the Earth.

We process one year of data products into a gridded dataset and then use this data set for further analysis, including inter-satellite comparisons, validation against AERONET, spatiotemporal data availability, LEO-GEO comparisons, and evaluation of the diurnal cycle at the regional and global scale.

### 4. 1 Inter-Sensor/Satellite Comparisons

We use the one-year, half-hourly (HH), 0.25° x 0.25° quarter-degree (QD), 30-minute gridded global data from three LEO and three GEO sensors to first intercompare the AOD from the individual sensors with each other. Figure 4 presents the density scatter plots comparing AODs from LEO and GEO sensors. The top row corresponds to AOD comparisons from MODIS-T with three GEOs. Similarly, the middle row is for MODIS-A, and the bottom row is for VIIRS-SNPP. We have selected AOD data for the first day of each month to manage the data volume for these inter-comparisons. Based on a small sample of comparisons with sunphotometer across different sensors,  AOD uncertainty for the DT algorithm (applied to MODIS) varies for land (±0.05 ±



15%AOD) and ocean (±0.03 ± 10%AOD), although actual numbers can vary by sensors due to differences in sensor characteristics and calibrations (Levy et al., 2010, Gupta et al., 2018/19). In Figure 4 we draw dotted lines to represent the ±(0.05+15%AOD) uncertainty envelope, which may best represent land retrievals while being generous to ocean retrievals. It is important to note that the aerosol regimes observed by the ABIs and AHI are very different. The ABIs cover the western Hemisphere with relatively

moderate AOD values (~0.1 to 0.5), whereas AHI covers Asia with high AODs (0.5 to 1.0).

Another way to compare sensor AOD is to look at regional mean values. If we consider each GEO-defined disk as a region, we can compare the mean AOD reported in the GEO-defined region for each individual sensor. The regional-mean technique combines differences resulting from collocated retrievals (captured in Figure 4) plus differences due to sampling by each individual sensor.

For plotting purposes we normalize the AODs of each individual sensor in each grid box by the values of the merged AOD in that grid box, and then calculate the daily regional mean normalized AOD within the GEO region. Figure 5 shows the timeseries of daily, regional, mean AOD normalized by the merged AODs for each GEO region. The normalized values closest to 1 indicate the primary contribution to merged regional mean AOD. Figure 5 shows that the GEO sensors in their respective region are the primary contributors to the merged AOD regional mean and datasets. The other sensors' deviation from the value of 1 is primarily due to

the relatively poor spatial-temporal sampling of the other sensors within the specific GEO-region. Note, only LEO sensors are plotted in Figure 5 for simplicity, although some overlap between GEO sensors exist. For the regional mean, MODIS-T is consistently higher than MODIS-A, with VIIRS-SNP in between. The differences among MODIS-T, MODIS-A, and VIIRS-SNP are consistent with previous studies (Levy et al. 2018, Lyapustin et al, 2023; Schutgens et al, 2020, Sogacheva et al., 2020) and represent retrieval differences that would be found in collocated comparisons and are not primarily due to sampling.

### 4.2 Spatial and Temporal Coverage

Figure 6 shows spatial coverage of the merged AOD product for each 30-minute gridded data file for a single UTC day, 26 March 2020 (Julian day number 86). The individual map shows the daylight portion of the earth for a given UTC time with color-coded

available AOD values mapped. In parts of the world with GEO sensors (Americas, East Asia), aerosol observations are available throughout the day under the cloud-free sky. The AOD data the from LEO sensors fill in some data gaps in these regions. The regions, including Africa, Europe, and part of Asia, have limited coverage provided by the LEO sensors alone. The high values of AOD over the Indian Ocean and Pacific are mainly due to pollution and smoke outflow from the continent. Southeast Asia also shows very high AODs associated with the active fire season.

These 30-minute files are then aggregated to calculate daily mean AODs for each grid and saved as a daily (UTC Day) global file. Daily means are simple averages of all available 30 minutes AODs for a given grid cell for a given day, no additional data filtering is applied. In addition to the average AOD value, other statistics include the number of 30-minute AOD values for the day, standard deviation, median, minimum, and maximum for each grid. The statistics can help further quality control the data suitable to address

specific research or/and application needs. Figure 7 shows an example of daily global coverage for March 26, 2020, from six individual sensors and merged datasets. Like 30-minute data, daily AODs from GEO sensors have more complete regional coverage due to high measurement frequency. However, LEO provides global coverage with data gaps due to less frequent (1 to few depending on latitude) measurements, additional data gaps come from cloudy, snow/ice surfaces, very bright surfaces, sun





glint over the ocean, and other DT retrieval limitations. The merged map (Figure 7) shows the daily mean AOD values for the global region with significant improvements in spatial coverage compared to any individual sensor.

We further quantify the spatiotemporal AOD data availability in the merged datasets and compare them with individual sensors. Figure 8 (top panel) shows a map of the average number of hours (per day) for which merged AOD data are available during the one-year study period. The average hours are calculated for each quarter-degree grid cell using merged datasets. The map is clearly divided into low temporal coverage areas observed only by LEO sensors (i.e., Africa and Europe) and high temporal coverage areas observed by both LEO and GEO sensors (i.e., Americas, Asia). For regions only covered by LEO sensors, AOD datasets are available only between 1-3 hours per day. The areas with GEO sensor coverage have 5.6±0.25, 5.6±0.23, 5.0±0.28 average (± one standard deviation) number of hours per day for ABI-G16, ABI-G17, AHI-H08, respectively. Data availability is highest for regions of GEO sensor overlap (i.e., part of the Atlantic Ocean, Western USA, and the Pacific Ocean). The regional variability in AOD data availability also depends on cloud/snow covers, length of daylight hours, and DT AOD retrieval limitations. For example, the white areas over the Saharan desert and Greenland are due to the DT algorithm not retrieving AODs over highly reflective surfaces (for any sensor). Another region with low data availability is around the InterTropical Convergence Zone (ITCZ), where there is a consistent cloud cover throughout the year.

Figure 8 (bottom panel) shows the day-to-day variability of the available number of grids (%) for daily mean AOD datasets, for six sensors and the merged product. The percentage of grids (y-axis) is calculated for each day and each sensor by dividing available grids with valid AOD data by the total possible quarter-degree grids covering the entire globe. The numbers next to sensor names in the figure legend are mean percentage grids for which AOD data are available on any given day. The GEO sensors (i.e., ABIs, AHI) provide about 13-14% daily spatial coverage at quarter degree resolution as their observations are limited to a specific region. These numbers are about 17-18% for two MODIS and 25% for VIIRS-SNPP due to its larger swath width and higher spatial resolution compared to MODIS. The advantage of merging six LEO and GEO sensors for the global AOD data set is evident in the percentage (44%) for the merged AOD dataset. Thus, it is safe to say that combining AODs from six sensors at quarter degree resolution can provide AOD datasets covering half of the globe, which is otherwise restricted to one-fourth or less from any individual sensor. Another interesting trend in the data availability is the seasonal cycle, in general, LEO sensors show peak coverage in the northern summer months (July-August) when the broad northern land mass is snow-free, whereas GEO sensors provide almost the same spatial coverage with very little month-to-month variability as they are more tropically biased and are mostly missing the Eurasian land mass . The merged AOD dataset's daily coverage varies between 40 and 50%, with a minimum in December and a maximum in August.

The daily mean AODs for individual sensors and merged products are then averaged over a month to calculate monthly AOD statistics for each quarter-degree grid cell. Figure 9 shows an example of monthly mean AOD maps for March 2020 for each sensor and merged product. The maps from each GEO sensor show their respective regional coverage, and the LEO sensor provides global coverage. The six sensors and merged products qualitatively represent the spatial distribution of aerosols consistently on a monthly scale. The merged product does have some discontinuities in transition regions. For example, AHI coverage is limited to half of India, and a clear line is visible in AOD maps separating AHI AODs from those obtained by merging AODs from LEO sensors. A similar arc is visible over the Atlantic Ocean, showing boundaries of ABI-G16 coverage. These sensor transition regions have differences in AODs and may need further quantification if research and application using merged AOD datasets are very sensitive



to minor errors. The transition and overlap area among GEO sensors are not apparent in monthly maps, and spatial distributions are homogeneous across sensors.

**4.3 Validation against AERONET**

The quarter-degree gridded AOD data from all six sensors and merged products are compared against measurements from AERONET. The spatiotemporal collocation of satellite AODs with AERONET over global locations is performed using standard validation practices [Ichoku et al., 2002, Gupta et al., 2018, Wei et al., 2020]. We followed the method detailed by [Gupta et al., 2020]. In brief, we locate the 0.25° grid box that contains the AERONET station and match the satellite-derived AOD for that grid box to the temporal average of all AERONET-measured AOD within ±15 minutes of satellite overpass time. This is done for each diurnal time step in the satellite data base. Thus, we are matching one gridded satellite AOD to the average of 1, 2 or 3 AERONET measurements, time step by time step. The collocated data are then used to perform inter-comparison analysis. The standard statistical parameters such as correlation coefficient (r), root mean square error (RMSE), mean bias (Bias), slope, intercept, and DT error envelope (EE= ± (0.05+ 15% x AOD)) are used to quantify the inter-comparison results (Table 3).

Figure 10 shows the inter-comparison of satellite and AERONET AOD for the one-year period over global locations. The three top density scatter plots are for LEO sensors, the bottom three are for GEO, and the central one is for the merged product. Table 3 provides the statistics corresponding to satellite versus AERONET AOD comparisons. The AODs from three LEO sensors each have a high correlation with AERONET (R = 0.83-0.88), whereas AOD from the GEO sensors have lower correlation (R = 0.66-0.82). For G17 the correlation is notably low (R = 0.655), but there are few cases of high AOD to define a linear signal. For the merged product, R=0.83. For the LEO sensors, MODIS-A and MODIS-T have similar RMSE, regression slope and EE%, but with an additional 0.01 bias for MODIS-T as expected (Levy et al., 2018). Also, as expected VIIRS-SNPP shows a larger bias with slightly increased RMSE and reduced EE% (Sawyer et al., 2020). Among GEO sensors, AHI has the largest mean bias (0.085) and RMSE (0.239), with the lowest EE value of 56.03%. The overestimation by AHI is also clearly visible in the density scatter plot. The results for AHI are consistent with our earlier study (Gupta et al., 2019), where we used two months (May-June 2016) of AHI data during KORUS-AQ field campaign to validate AHI AODs with AERONET. In Gupta et al., 2019, we reported the mean bias (0.09) and RMSE (0.20), EE% (55%), R(0.84). The MODIS validation during same period in the region with AHI coverage also demonstrate similar validation statistics (Gupta et al., 2019). The current DT algorithm faces challenges in the Asian region due to complex and varying sufaces, and highly varying aerosol chemical composition. The LEO sensor's validation statistics are consistent with those performed using level 2 aerosol products [Sawyer et al., 2020; Wei et al., 2020, Levy et al., 2018, Gupta et al., 2020]. The merged AOD has a mean positive bias of 0.051 with a slope value of 1.1 and an EE value of 65.45%, mainly driven by the GEO retrievals.

Figure 11 presents the spatial distribution of correlation coefficient (top left), EE % (bottom left), mean bias (top right), and RMSE (bottom right) calculated from the MERGED-AERONET collocation datasets at individual AERONET stations. The regional distribution of validation results can be critical for local and region-specific data applications. It is also important to note that atmospheric aerosols, including their types and sources, are highly variable in space and time. In addition, DT AODs have shown variability in uncertainties in different parts of the world due to changes in surface reflectance, topography, and varying aerosol properties. The correlation varies between 0.33 and 0.98, with 60% of stations showing a value greater than 0.8. Only about 5% of



stations have a correlation value of 0.5 or smaller (black dots in Figure 11a). The low value of correlation is typically visible in the western US, parts of South America, and some isolated stations in Asia. These results are consistent with previous validation studies [Levy et al., 2018, Wei et al., 2020, Gupta et al., 2019, Sawyer et al., 2020]. There is a string of stations that begin in the Indo-Gangetic Plain and extend southeastward to Southeast Asia. These stations show overall poor validation statistics, which

drive the poor showing of AHI in Table 2. In general, the EE% is greater than 80% for 29% of stations, and 58% of stations have a value greater than 67%.

Figure 12 shows the global and seasonal mean diurnal cycle of AODs from AERONET and the merged product. The collocated datasets were used for diurnal cycle analysis. AODs from all the AERONET stations for each hour (in local solar time or LST)

and for the season are averaged to generate these plots. We choose to use LST instead of UTC time to understand the patterns in AODs from morning to evening hours globally while keeping solar geometry similar. The AERONET AOD patterns are very similar in all seasons, with the peak around local noon and lower values in the morning and evening hours. These patterns in AERONET AODs are more prominent in northern hemispheric spring (AMJ) and summer (JAS), where the magnitude of the diurnal signature increases by 50% and 15%, respectively, from morning to midday. The AERONET diurnal AOD signatures are

flatter in fall (OND) and winter (JFM). In all four seasons, merged AODs show positive bias against AERONET AODs, as expected from the scatterplots of Figure 10. However, despite the bias, the merged satellite AODs follow the AERONET diurnal trend throughout the day except during early morning and late afternoon hours. The mismatch during those hours can be related to sampling inconsistency depending on AERONET stations' locations. It is important to note that the number of AOD samples used to get the mean AOD for each 30-minute interval varies significantly (Secondary y-axis). The sampling is highest around local

noon and significantly lower at the beginning and end of the day. Thus, we calculate the mean bias for each season while considering hours with only significant sampling (at least 1000 pairs, white dotted points in Figure 12). The mean bias varies between 0.03-0.06, with the highest value in spring (AMJ) & summer (JAS) and the lowest in fall and winter.

## 5   Summary and Conclusion

For two decades, LEO sensors (MODIS on Aqua, MODIS on Terra and VIIRS on Suomi-NPP) have been a source of high-quality aerosol observations with moderate spatial resolution from space. More recently, GEO sensors have been making similar observations with the launch of the AHI in 2014 aboard Himawari-8 and then continued by Himawari-9, GOES-16, GOES-17, and, more recently, GOES-18. The LEO sensors (i.e., MODIS, VIIRS) can provide 1-2 measurements per day across the globe,

but GEO sensors provide high frequency (~10 minutes) data for Full-Disk imagery over a particular region.

In this study, we implemented the well-known Dark Target aerosol retrieval algorithm on three LEO and three GEO sensors and processed one year of Level 2 aerosol products from all sensors. The level 2 aerosol optical depth at 550 nm from each sensor was then gridded into a quarter-degree grid for every 30 minutes of observations. The gridded individual sensors' AODs were then

merged by averaging available AODs in each quarter-degree grid cell for a given 30-minute time window. This way, we have created a gridded AOD product with spatial and temporal resolutions of a quarter degree and half an hour, respectively. The final gridded AODs from individual sensors and merged datasets are saved into a 30-minute global file, thus making 48 files per day. The 30-minute files were then used to estimate the daily mean AODs, and daily AODs were used to calculate monthly mean AOD





datasets. This is the first moderate resolution gridded AOD data set merged from six separate sensors available globally at a temporal resolution that can discern diurnal signatures.

The gridded data sets from individual sensors have been compared against each other, validated against ground measurements over global locations, and errors are characterized. The merged product has a global mean bias of about +0.05, with 65% of retrievals falling within ± (0.05+ 15% x AOD), with the majority of retrievals on the positive-biased side of the stated error bars .

The merge provides an excellent global coverage with a high frequency of AOD measurements in the regions covered by GEO sensors (Americas & Asia). There are temporal data gaps in regions with no equivalent GEO sensors (i.e., Europe and Africa). The
10 LEO sensors fill in some of the spatial data gaps in the GEO sensor's field of view due to cloud or retrieval limitations such as viewing angles and sun glint. The merged product provides almost 45% global coverage on any given day, which is a significant improvement to the 12-30% expected from any individual sensor used in this study.

Such merged gridded data can assist in tracking the aerosol transport resulting from wildfires and dust storms, allowing for better
air quality forecasting and hindcasting. The high spatiotemporal resolution of new AOD datasets will help evaluate and inter-compare regional or global model simulations and reanalysis products in ways previously unattainable in a gridded format.

The merged product is able to approximate the diurnal cycle of AERONET AODs, although with a positive bias. We note an unexpected strong diurnal signature from the global composite of AERONET AOD during northern Spring. In this global
composite mean AERONET AOD begins the morning with a value of 0.12, reaches midday with a value of 0.19, which drops back down to 0.12 at sunset. That is a diurnal signal of 50%. Previous studies using AERONET to determine diurnal signal for AOD find diurnal signatures of that magnitude for individual sites or groups of similar sites, but when aggregated into larger composites of multiple disconnected stations those signatures flatten (Kaufman et al., 2000, 2005, Smirnov et al. 2002, Zhang et al., 2012, Arola et al. 2013).

The Level 2 and merged Level 3 data presented in this study are prototypes of products proposed in response to NASA's Making Earth System Data Records for Use in Research Environments (MEaSUREs) Program. The intention is to use a version of retrieval code, known as the Dark Target Package (DT-Package), that further homogenizes the Level 2 retrievals from the three LEO and three GEO sensors and leads to improved consistency. Overall, this effort will generate a "Version 1.0" product. We expect to see
minor differences to retrieved AOD as well as diagnostics and quality flags, which should not greatly impact the statistics presented. On the other hand, there will be great improvements to file structures and file metadata, as well as processing robustness, enabling these data to be archived at NASA's LAADS DAAC. Processing will cover the years 2019 – 2022.

The intended MEaSUREs Level 3 products will also include some updates, both to file usability as well as content. In addition to
35 the best quality AOD (the filtered "Optical_Depth_Land_And_Ocean") at 550 nm, these Level 3 data will also include statistics based all-quality AOD data (non filtered "Image_Optical_Depth_Land_And_Ocean"). Thus a user will be able to consider an ensemble approach to estimating AOD. In addition, we will also provide QDG sun-satellite geometries for each sensor separately. The Level 3 data file will also have AOD values from each sensor at QDG along with merged AODs.



It is expected that the Version 1.0 delivery of the GEO-LEO dataset will still be impacted by most of the same offsets and biases (compared to each other, and compared to AERONET) as presented here. However, the adaptation of the unified DT-Package will enable systematic improvement. For example, Kim et al., (https://doi.org/10.5194/amt-2023-128) is applying a new surface reflectance parameterization to the GEO retrievals that better accounts for the different GEO versus LEO observing

5    geometry. This will help address some of the remaining diurnal biases, which will then help characterize aerosol effect on radiative forcing, cloud development and air quality. We expect the merged satellite data set presented here, and its future updates will be a major asset in exploring the sources and consequences of aerosols' diurnal variability.

## 6    Acknowledgment

This work was supported by NASA's EOS program and the NASA ROSES program NNH17ZDA001N: Making Earth System Data Records for Use in Research Environments (MEaSUREs) managed by Lucia Tsaoussi. We thank Space Science and Engineering Center (SSEC), University of Wisconsin-Madison for providing Himawari-8 data, and to NASA's MODIS Adaptive Processing System (MODAPS) Services for providing access to NOAA's GOES-16 and -17 data. We thank the MODIS and VIIRS

Calibration and Support Teams MCST/VCST for their efforts to maintain and improve the radiometric quality of MODIS and VIIRS data, and MODAPS and LAADS for the continued processing and archive of the MODIS products. The AERONET team (GSFC and site PIs) are thanked for the creation and continued stewardship of the sun photometer data record; which is available from http://aeronet.gsfc.nasa.gov.

**Data availability**

The aerosol datasets from six sensors produced using the dark target algorithm have been as part of NASA's MEaSUREs project (ROSES-2017; https://www.earthdata.nasa.gov/esds/competitive-programs/measures/leo-geo-synergy), and are publicly available at NASA LAADS (https://ladsweb.modaps.eosdis.nasa.gov/archive/allData/5019). The AERONET direct sun measurement data

used in this study are available via the AERONET website (https://aeronet.gsfc.nasa.gov/new_web/download_all_v3_aod.html).

**Author contributions**

PG, RL, and LR designed this study. SM, PG, ZZ, and WS developed codes and carried out the level 2 retrievals; PG developed

the level 3 data processing algorithm and ran the experiment and analysis. SZ, PG, JW, XP, RL, MO, and VPK developed and tested level 3 operational code. PG, LR, and RL prepared the manuscript draft, and all authors reviewed it.

**Competing interests**

The contact author has declared that none of the authors has competing interests.

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

35





Table 1: Level 2 aerosols scienfitic data sets used in the study

| Geolocation/Geophysical Parameters | Description |
|---|---|
| Latitude | Latitude of the center of Level 2 pixel |
| Longitude | Latitude of the center of Level 2 pixel |
| Sensor_Zenith | Satellite/Sensor Viewing Angle |
| Solar_Zenith | Solar Zenith Angle |
| Scattering Angle | Scattering Angle |
| Optical_Depth_Land_And_Ocean | AOD at 550 nm over land and ocean with recommended high quality flags only. |

35

40





Table 2. Satellite/Sensor used in this study and their spectral bands, resolution, coverage, data version, and other characteristics. The wavelength/resolution are in μm, km respectively. The '-' represent missing band in a particular sensor.

| Characteristics | MODIS-T | MODIS-A | VIIRS-SNPP | ABI-G16 | ABI-G17 | AHI-H08 |
|---|---|---|---|---|---|---|
| Blue | 0.47/0.50 | 0.47/0.50 | 0.49/0.75 | 0.47/1.0 | 0.47/1.0 | 0.47/1.0 |
| Green | 0.55/0.50 | 0.55/0.50 | 0.55/0.75 | - | - | 0.51/1.0 |
| Red | 0.65/0.25 | 0.65/0.25 | 0.67/0.75 | 0.64/0.5 | 0.64/0.5 | 0.64/0.5 |
| NIR | 0.86/0.25 | 0.86/0.25 | 0.86/0.75 | 0.86/1.0 | 0.86/1.0 | 0.86/1.0 |
| SWIR | 1.24/0.50 | 1.24/0.50 | 1.24/0.75 | - | - | - |
| Cirrus | 1.38/0.50 | 1.38/0.50 | 1.38/0.75 | 1.37/2.0 | 1.37/2.0 | - |
| SWIR | 1.64/0.50 | 1.64/0.50 | 1.61/0.75 | 1.60/1.0 | 1.60/1.0 | 1.61/2.0 |
| SWIR | 2.11/0.50 | 2.11/0.50 | 2.25/0.75 | 2.26/2.0 | 2.26/2.0 | 2.25/2.0 |
| Level 2 Data Resolution | 10 km | 10 km | 6 km | 10 km | 10 km | 10 km |
| Land/Sea Mask Resolution | 1km | 1km | 750 m | $0.01^0$ | $0.01^0$ | $0.01^0$ |
| Level 2 file length (Minutes) | 5 | 5 | 6 | 10 | 10 | 10 |
| Coverage | Global | Global | Global | Americas-Atlantic | Americas-Pacific | Asia-Pacific |
| Operational Data Version | C6.1 | C6.1 | V2.0 | Beta (V0) | Beta(V0) | Beta(V0) |
| Product Name | MOD04_L2 | MYD04_L2 | AERDT_L2_VIIRS_SNPP | AERDT_L2_ABI_G16 | AERDT_L2_ABI_G17 | AERDT_L2_AHI_H08 |



Table 3. Summary statistics of satellite-AERONET comparison for the global region

| Sensor-Satellite | Number of Pairs | EE% | Bias | RMSE | R | Slope | Intercept |
|---|---|---|---|---|---|---|---|
| MODIS-T | 17208 | 68.00 | 0.020 | 0.116 | 0.873 | 0.96 | 0.020 |
| MODIS-A | 15359 | 70.76 | 0.011 | 0.110 | 0.876 | 0.99 | 0.004 |
| VIIRS-SNPP | 32931 | 62.38 | 0.050 | 0.149 | 0.825 | 1.09 | 0.019 |
| ABI-G16 | 141351 | 66.00 | 0.048 | 0.108 | 0.791 | 1.12 | 0.021 |
| ABI-G17 | 79079 | 72.29 | 0.031 | 0.081 | 0.655 | 0.92 | 0.023 |
| AHI-H08 | 61630 | 56.03 | 0.085 | 0.239 | 0.819 | 1.13 | 0.022 |
| MERGED | 272725 | 65.45 | 0.051 | 0.147 | 0.833 | 1.10 | 0.020 |



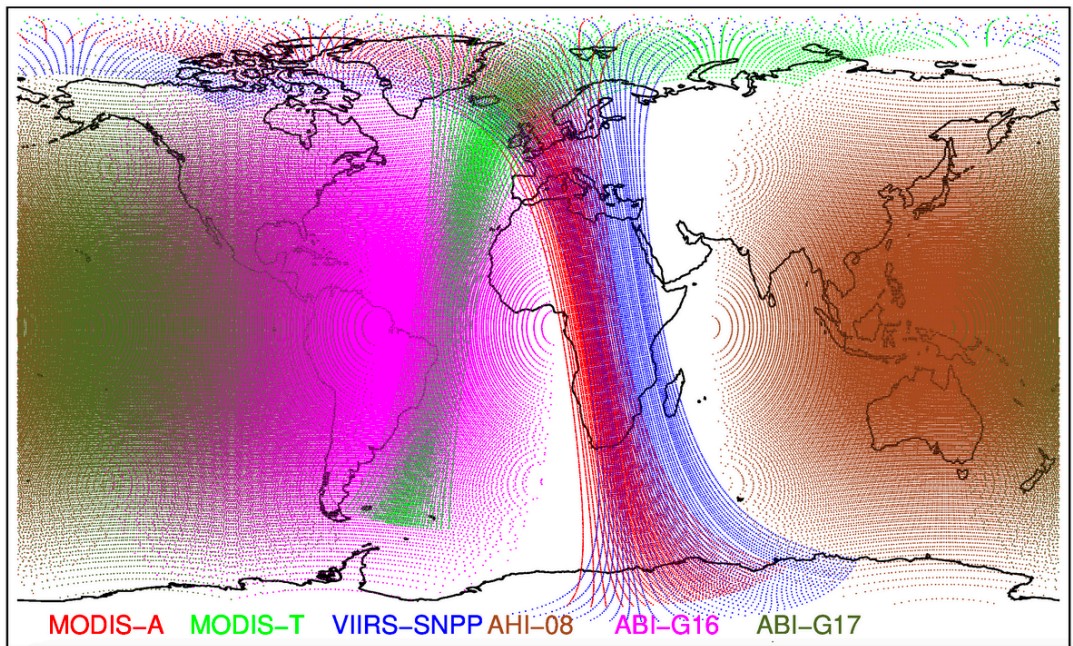

Figure 1. An example of spatial coverage by six sensors used in the study using level 2 data. The data from September 7, 2019, for varying time periods from each sensor are plotted. The map only represents every 5th pixel from two MODIS, every 10th pixel from VIIRS, and every 3rd pixel from GEO sensors. The pixel omission is done to ensure visibility of each sensor's coverage on the map. We choose to show one orbit of LEO sensors and one full disc image of GEO sensors. The purpose of the map is schematic to show relative coverage by each sensor.



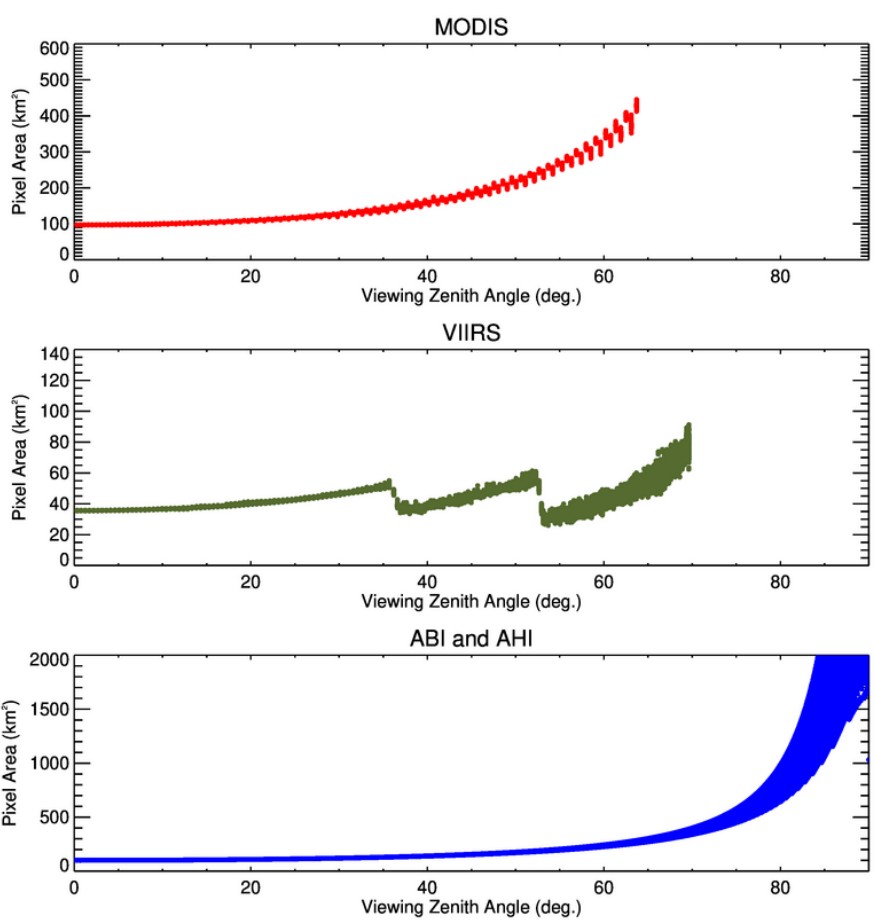

Figure 2. Pixel area (km²) of level 2 data as a function of sensor viewing geometry for all six sensors. The corresponding maps (Figure 1) show the locations of the individual swath. The actual pixel area for LEO and GEO sensors is expected to vary in different parts of the world as a function of latitude. The figure used the same data as visualized in Figure 1.





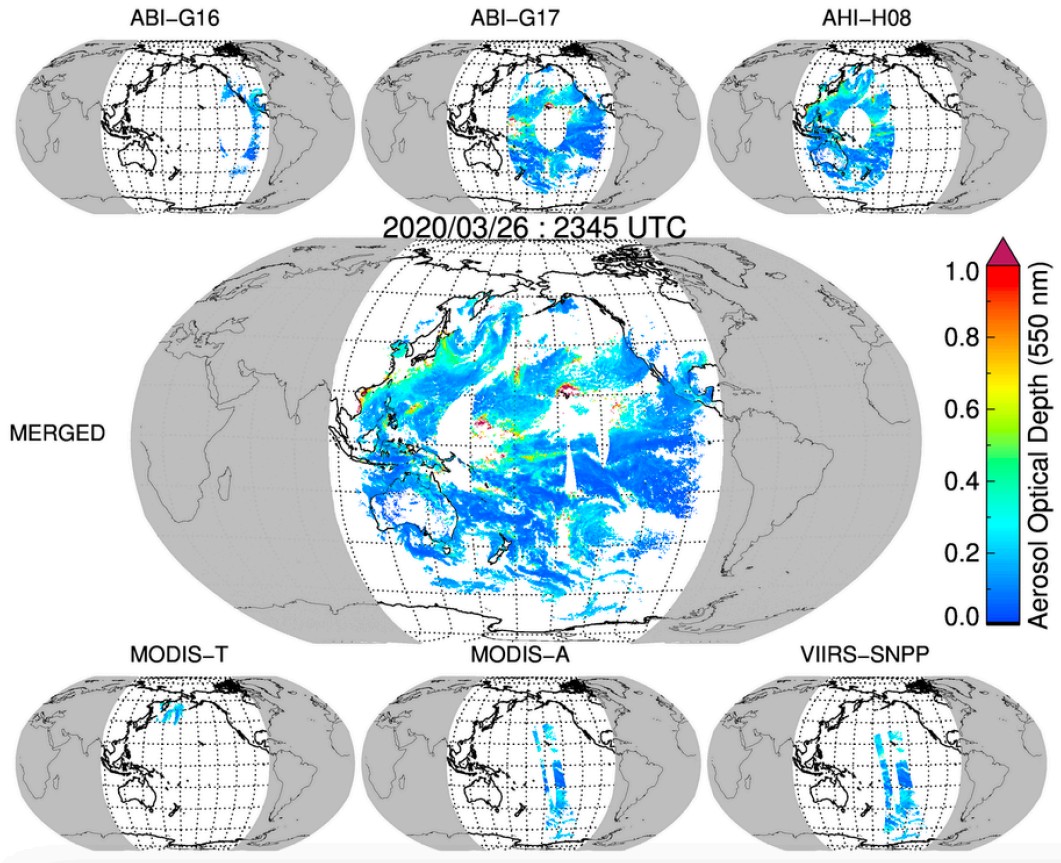

Figure 3. Example of 30 minutes aerosol optical depth data coverage by each sensor and merged datasets on March 26, 2020, at 2345 UTC.

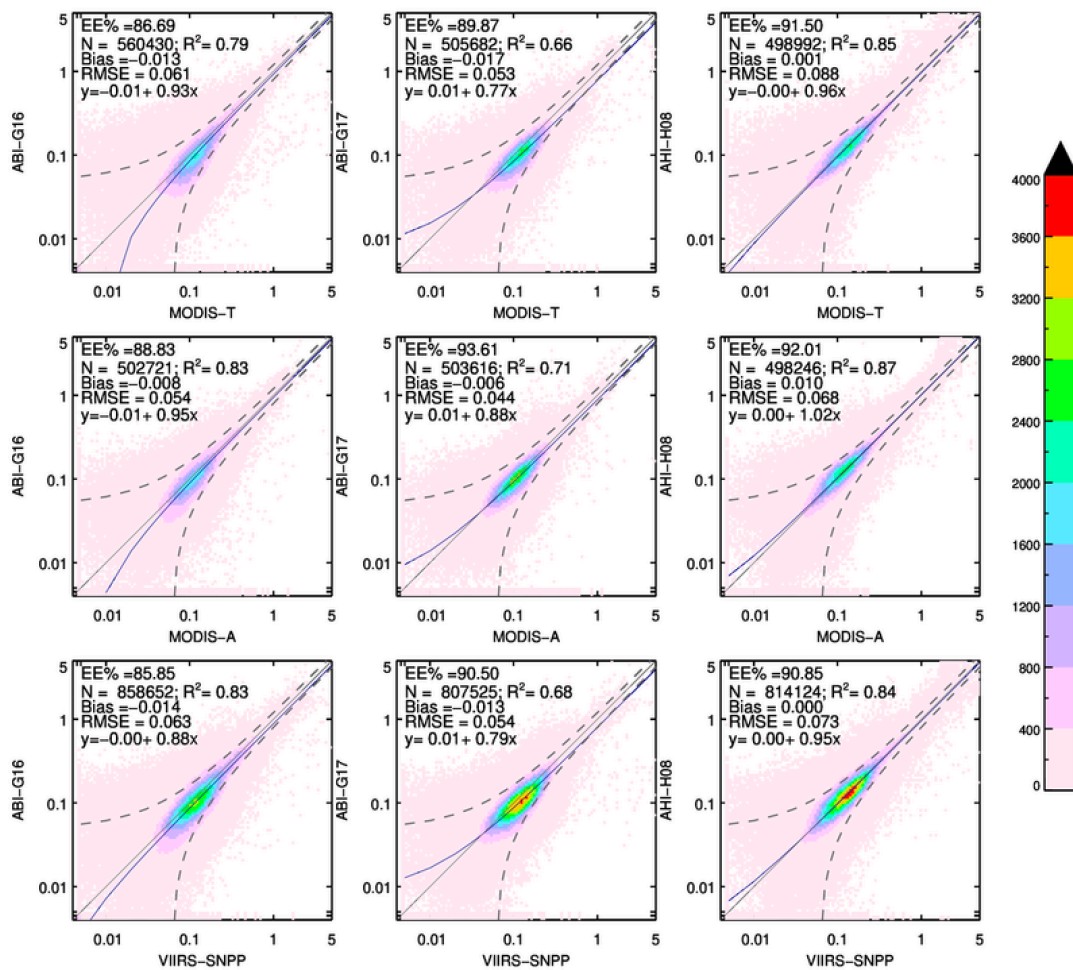

Figure 4. The inter-comparison of AOD (550 nm) from three LEO and three GEO sensors. The QD gridded AOD data from LEO and GEO sensor for one day of each month of the year are selected and used to inter-compare coincident LEO and GEO AOD retrievals.

Figure 5. The ratio of regional daily mean AOD for each sensor and coincident merged AODs. The three plot shows data time series from three GEO regions. The black line is the ratio of GEO AODs to merge whereas red, blue, and green represent the ratio of MODIS-T, MODIS-A, and VIIRS-SNPP with merged AODs respectively.





Figure 6. Example of 30 min coverage of merged aerosol data for one entire day (March 26, 2020, UTC). Each panel shows day light portion of earth at specific time (UTC) and available AOD retrieval from the six sensors merged product.





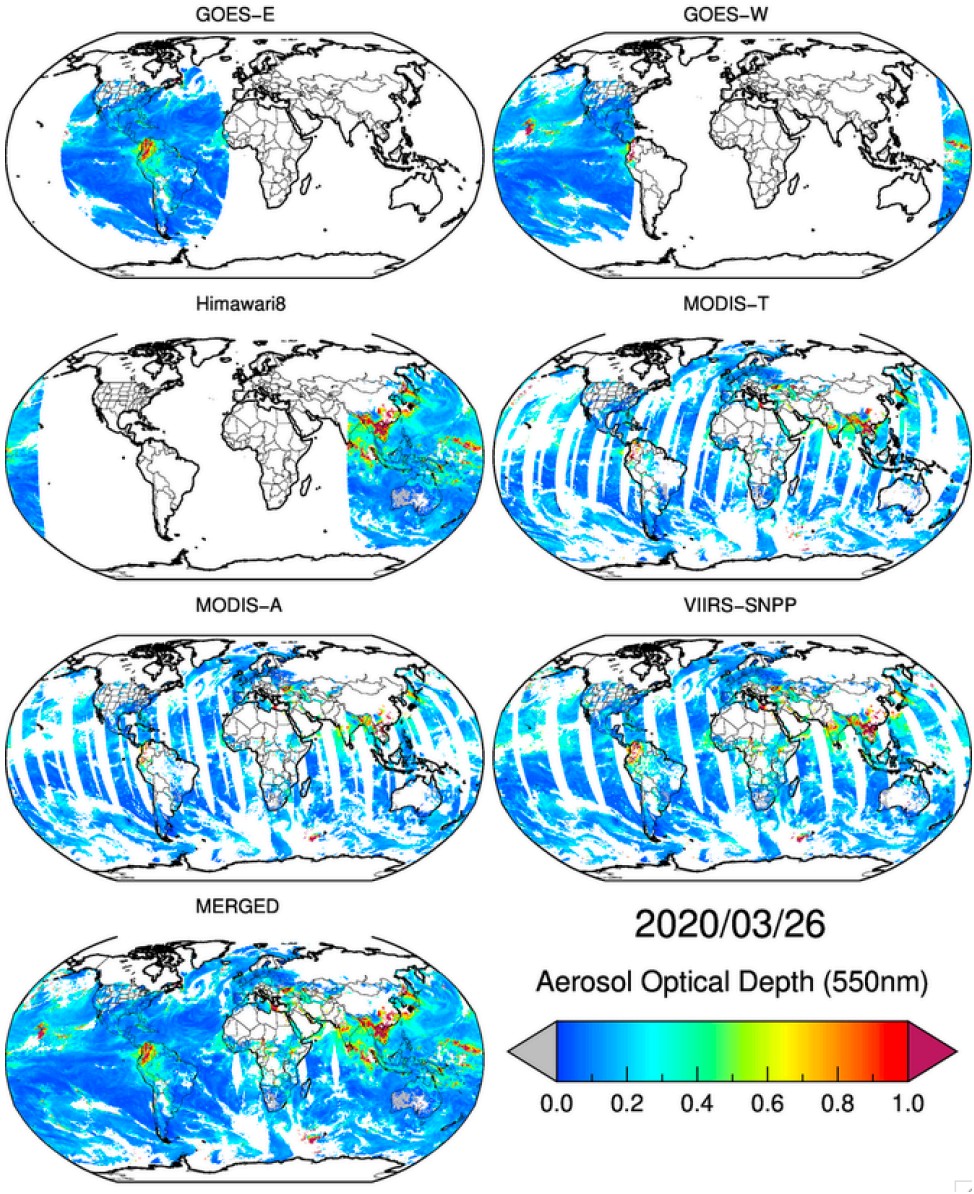

Figure 7. Daily average aerosol optical depth (550 nm) on March 26, 2020, retrieved by six LEO and GEO sensors and merged quarter degree product. The daily averages (daytime) are calculated using 48 of 30 minutes, quarter-degree gridded files.



Figure 8. Spatial and temporal AOD data availability for April 2019 to March 2020 using merged datasets. The top map is mean number of hours in a year for each grid. The bottom panel is number of quarter degree grids (%) where daily average AOD data are available from six sensors and in the merged dataset.







Figure 9. Monthly average aerosol optical depth (550 nm) for March, 2020, retrieved by six LEO and GEO sensors and merged quarter degree product. The monthly averages (daytime) are calculated using daily average AOD values (shown in Fig 7), quarter-degree gridded files.



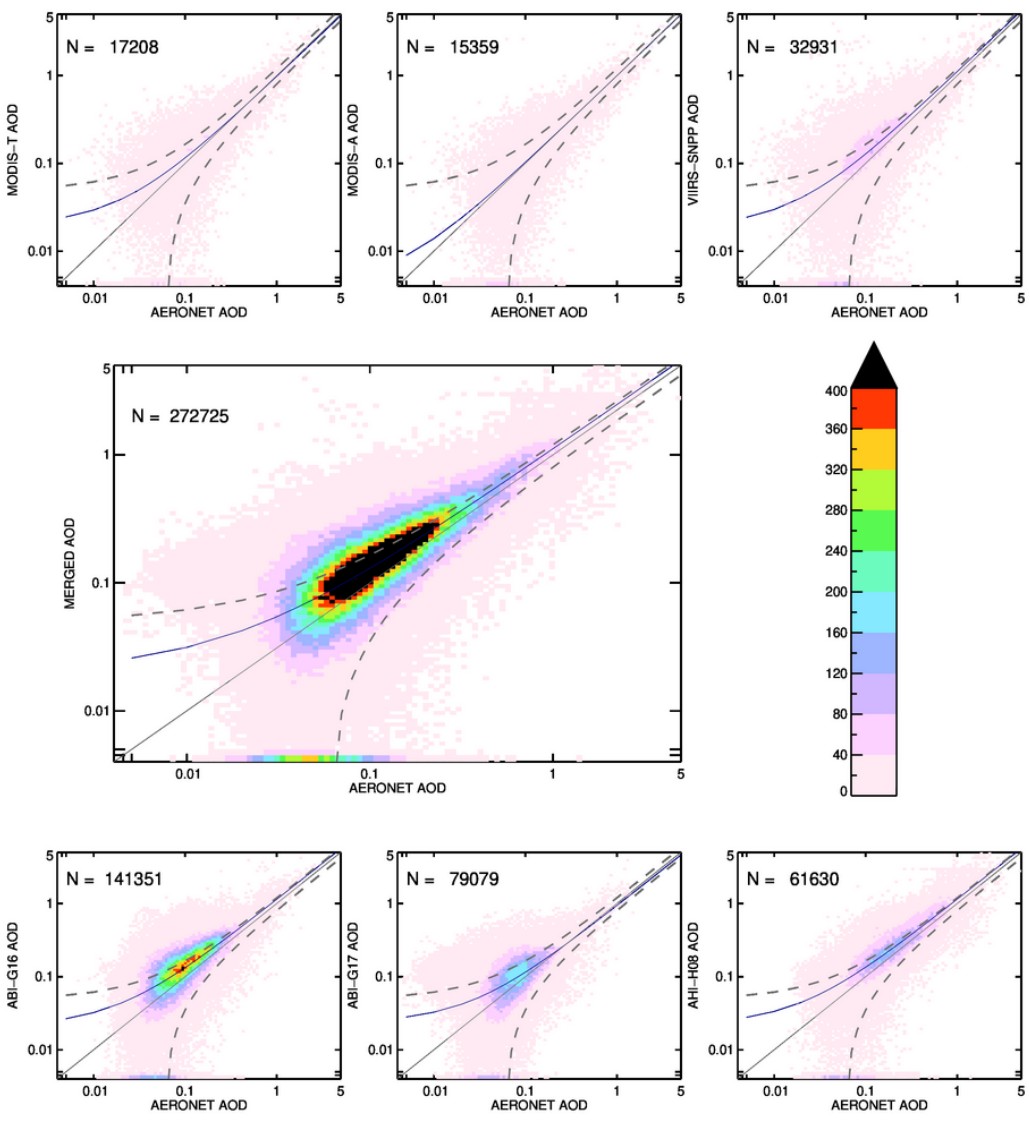

Figure 10. Validation of Quarter-Degree-Grid (QDG) AODs from six sensors and merged datasets with AERONET measurements across the globe. The spatial match is done by picking nearest QDG to AERONET location and temporally AERONET AODs are averaged for ±15 minutes around satellite time. Top row, left to right, MODIS-T, MODIS-A, VIIRS-SNPP. Middle: merged product. Bottom row, left to right, ABI-G16, ABI-G17, AHI-H08.



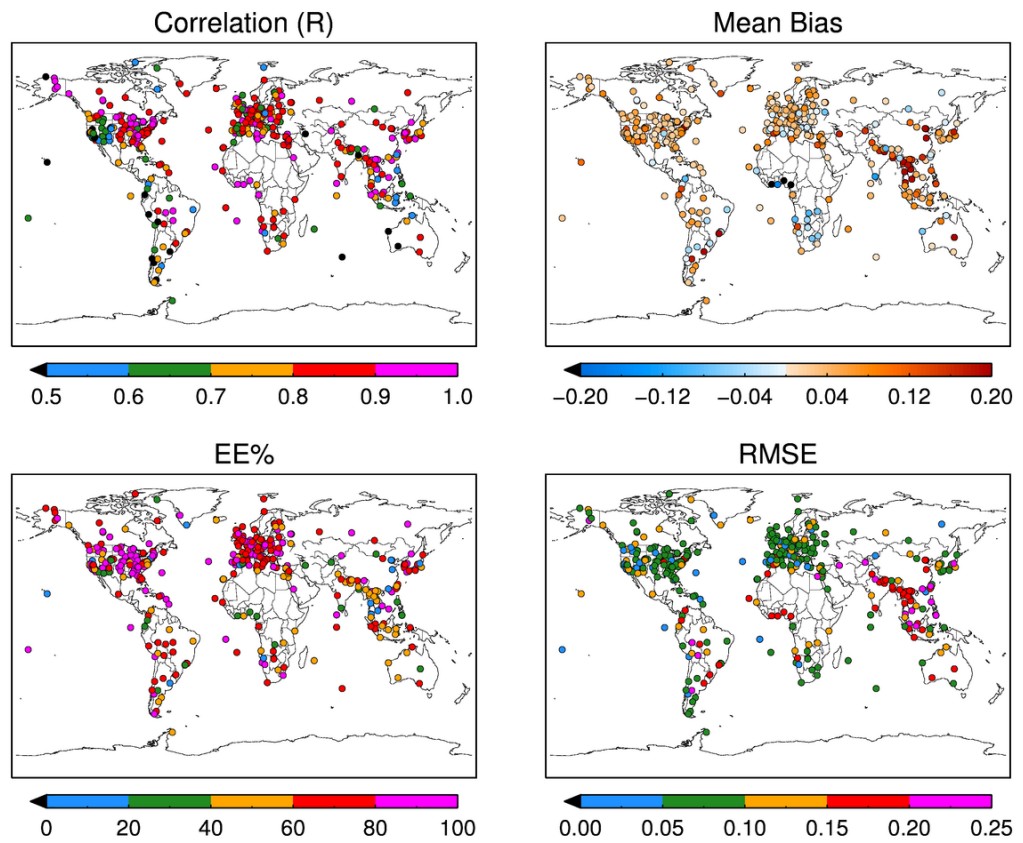

Figure 11. Summary statistics of satellite-AERONET comparison for individual stations across the globe. The panels are correlation coefficient (top left), mean bias (top right), bottom left (EE%), and bottom right (RMSE).



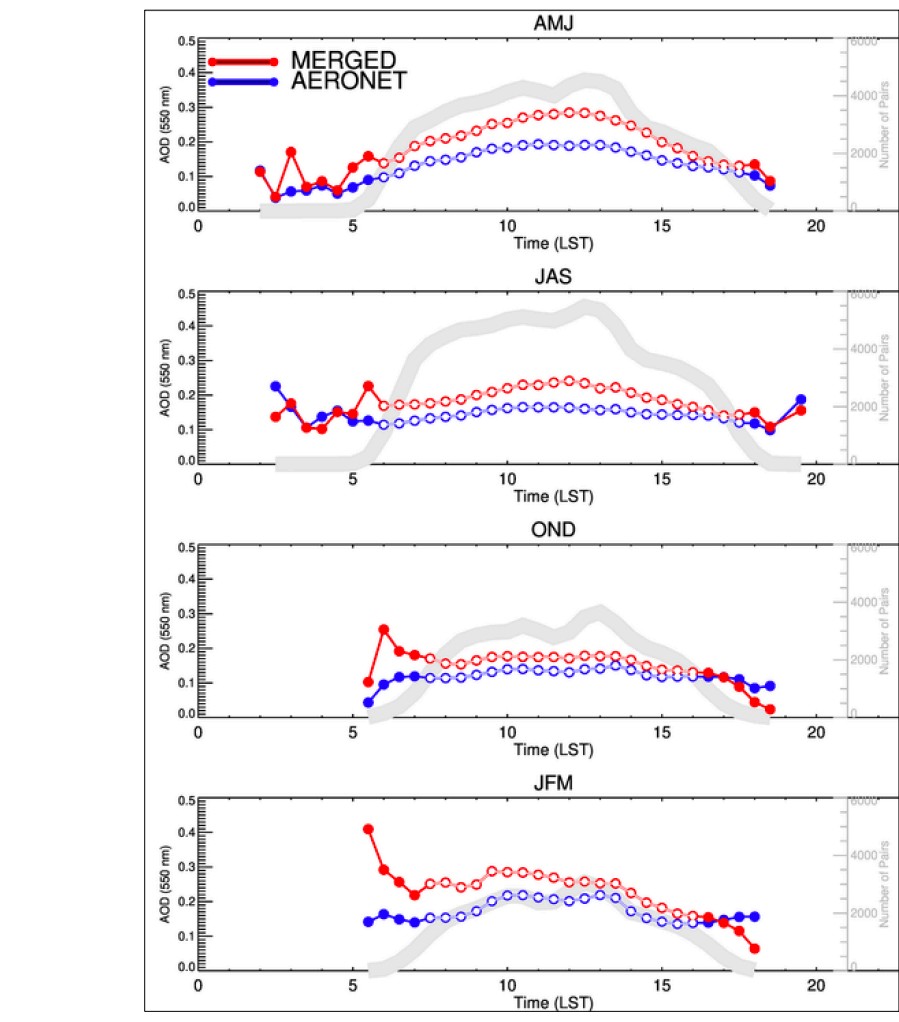

Figure 12. global diurnal cycle in local solar time as a function of season (. The open circle shows hours with a minimum of 1000 samples to average. The number of samples used to get the average for every 30 minutes of data is shown on the secondary y-axis. The solid circles in the morning and evening hours represent lower sampling and demonstrate more variability. Each panel repsents a seaons, AMJ: April-May-June, JAS: July-August-September, and OND: October-Number-December, JFM: January-February-March. These are bit different than most meteorgical seaonal definition.
