# Peer review of "Increasing Aerosol Optical Depth Spatial And Temporal Availability By Merging Datasets from Geostationary And Sun-Synchronous Satellites"

_Atmospheric Measurement Techniques, 2023_

## Author Comment (AC1)

**RC2: 'Comment on amt-2023-259', Anonymous Referee #2**

**General Comments:** (original comments are in black and responses are in blue)

This manuscript presents advances in the monitorization of aerosol optical depth from satellite measurements by increasing the temporal resolutions at 0.25º grid. That permits studies of aerosol evolution during the day. To do so, the authors apply the Dark Target algorithm to six different satellites, three Low Earth Orbit (LEO) and three with Geostationary Earth Orbit (GEO). They also proposed a new product that merges data from the different sensors. Authors also present intercomparisons between the different sensors, validation versus AERONET and eventually the diurnal AOD cycle from satellite observations. I believe that all these issues are very interesting and will permit advances in aerosol studies. However, in the current state the manuscript requires revision before its publication in AMT.

We thank the reviewers for insightful comments that have improved the manuscript significantly. In particular, we realized that there was room for additional examples of aerosol diurnal signatures and have thus added some. Furthermore, we also realized that our two goals of creating this data set were not apparent to the reviewers and have adjusted wording in several place to explicitly state our two goals and to clarify that we are merging already existing aerosol products, not creating a new algorithm from scratch. The goal here is a ready-to-use collocated data set on a global grid. We do not expect this data set to be any more accurate than the data that feeds into it, but we do expect it cover more **spatial** and **temporal** territory than any single sensor can provide.

Again, our thanks to the reviewers for helping us explain the study better.

To me the methodology for the creation of the new merged AOD product is not clear. The methodology proposed seems not to exploit the potential of combining all measurements in a single retrieval. Indeed it seems that it is just a statistical approach combining the same retrieval (dark target) on different sensors. I recommend the authors to further explain the new methodology. For example, box-diagrams for caveat 1 and caveat 2 in section 3.3 could help to further understanding.

Added the term "level 3" into the abstract, which is internal jargon that we were trying to avoid but is a clear indication of what the data set is all about.

Added text in the last paragraph of the Introduction to bound the scope of the study.

"This data set addresses two goals. First, it increases spatial coverage by filling in the holes caused by clouds and glint in the global aerosol product. Second, the data set fills in the temporal holes in the diurnal signature caused by the once-per-day sampling of the polar orbiting sensors. It is this synthesis of spatially- and temporally-rich data and the uniqueness of a fine resolution global grid, easily ingested by global models and gridded data systems that offers new value to the aerosol community."

Added text to the last paragraph of Section 2 to further clarify the difference between Level 2 & Level 3

We have added a sentence to the end of the 1st paragraph of Section 3.1 to be specific of the starting point of our aggregation. We start from level 2, not from the measured reflectances.

We added text to the first paragraph of Section 3.3 to reiterate that we are starting from Level 2 AODs and not from Level 1 spectral radiances.

With these additional text insertions, the reader is now guided to a better understanding of the methodology and will not be taken by surprise that the gridding process is a statistical approach combining AOD products and not a fresh retrieval using combined Level 1 radiances. We do not see a way to create a diagram that would help further. We refer to our earlier study (Gupta et al., 2020), which further explains with graphics.

Caveat 2 explores future improvement possibilities while providing weight to each retrieval based on its inherent retrieval quality. Unfortunately, in the current version of the DT algorithm, we don't have an uncertainty parameter that can be used to weight the retrieval. This step required further research and analysis.

I believe that the current analyses of diurnal cycles are poor. Authors present the diurnal cycle for all possible AERONET stations and no remarkable diurnal cycle is observed. In previous AERONET studies (see Smirnov et al., 2002) diurnal cycle is site-dependent due to the different aerosol regime in each station. Also, in Smirnov et al., 2002 diurnal cycles are studied in percentage deviation versus daily means, that allow to identify diurnal patterns. Following the same approach could serve to see the potentiality of increasing the number of daily measurements in satellite observations. Studies cases would give clarity.

We agree with the reviewer, added some examples of diurnal cycles on specific AERONET stations, and provided additional figures in the supplementary material. At this point, we are unable to provide a percentage deviation as suggested. This required significant additional data analysis, including the AERONET and satellite data sampling biases. In fact, we are working with Smirnov and their team to revise and update their 2002 analysis using a newer version and a more extended period of datasets.

Apart of the two major concerns, I also have other minor concerns:

Introduction:

Page 1, Lines 25-27: This claim about diurnal cycles in AOD does not correspond with the analyses in the manuscript (see major comment).

We added more examples of diurnal cycles to expand on Figure 12 analysis and revised the text to support scientific conclusions further.

Page 2, Lines 4-5: The statement that there is information on aerosol microphysical properties from Dark Target is not correct. To my knowledge Dark Target does not provide aerosol microphysics as an output. Indeed, the algorithm assumes predefined aerosol models and uses look-up tables.

The reviewer is correct, and we do not use the term "microphysics". However, over the ocean specifically, the retrieval is sensitive to particle size. The pre-defined aerosol models in the look-up tables are single size modes. The algorithm matches one fine mode with one coarse mode and determines the relative weight of each. The retrieved solution to the over ocean retrieval is the spectral AOD from visible to SWIR, itself a measure of particle size, and a parameter that we call fine mode fraction that gives this relative weight between the chosen modes in the look-up table. The fine mode fraction from the ocean retrieval is well-used in multiple studies (Yan et al., 2022; Levy et al., 2013; Vinoj et al., 2014; Bellouin et al., 2005). Over land, the process is different; the range of the retrieved spectral AOD is only within the visible wavelengths, and the choice of aerosol model in the look-up table is much restricted, although even over land, the retrieval is given the opportunity to weight two bimodal models, one fine mode dominated and the other coarse mode dominated.

Yan, X., Zang, Z., Li, Z., Luo, N., Zuo, C., Jiang, Y., Li, D., Guo, Y., Zhao, W., Shi, W., and Cribb, M.: A global land aerosol fine-mode fraction dataset (2001–2020) retrieved from MODIS using hybrid physical and deep learning approaches, Earth Syst. Sci. Data, 14, 1193–1213, https://doi.org/10.5194/essd-14-1193-2022, 2022.

Bellouin, N., Boucher, O., Haywood, J., and Reddy, M. S.: Global estimate of aerosol direct radiative forcing from satellite measurements, Nature, 438, 1138–1141, https://doi.org/10.1038/nature04348, 2005.

Levy, R. C., Mattoo, S., Munchak, L. A., Remer, L. A., Sayer, A. M., Patadia, F., and Hsu, N. C.: The Collection 6 MODIS aerosol products over land and ocean, Atmos. Meas. Tech., 6, 2989–3034, https://doi.org/10.5194/amt-6-2989-2013, 2013.

Vinoj, V., Rasch, P. J., Wang, H., Yoon, J.-H., Ma, P.-L., Landu, K., and Singh, B.: Short-term modulation of Indian summer monsoon rainfall by West Asian dust, Nat. Geosci., 7, 308–313, https://doi.org/10.1038/ngeo2107, 2014.

Page 2, Lines 19-20:  It is trivial that geostationary observations have more chance of getting measurements. It needs to be re-phrase

Text will be revised as

"In fact, because a scene is rarely continually cloudy from sunrise to sunset, we see that geostationary sampling can find at least one cloud-free opportunity to make an aerosol retrieval on any day, This increases the probability of at least one aerosol retrieval sometime during the day to nearly 100%  (Remer et al., 2012)."

Page 2, Line 19: Again, the larger temporal resolution of geostationary observations is straightforward.

Agreed, text revised as above.

Page 2, Lines 26-27: It is straightforward the presence of aerosol diurnal variability from any ground observations. Why not remarking the limitations of LOE observations?

We added a sentence describing the failure of polar-orbiting sensors to fill in temporal holes and resolve diurnal cycles on a global basis. We also provide an example of a diurnal cycle at a location where current coverage is only by LEO sensors. This is part of the newly added diurnal cycle analysis.

Page 2, Lines 33-34: I guess that authors want to mention that studies of AOD diurnal patterns have been made using AERONET observations.

Yes, Sentence added with references.

Page 3, Lines 4-6: Why covering the globe with multiple geostationary sensors is not the same as viewing the entire globe with MODIS-like sensors? I do not get the point. Could this be related with the different spatial resolutions? Also, why having two identical sensors do not provide the same aerosol results unlike MODIS?

Instrument calibration and characterization is an extremely challenging activity, and long-term consistency in products derived from multiple sensors is even more challenging. Even two identical MODIS instruments do not produce the same values of AOD (Levy et al. 2018). Without vicarious calibration and inter-sensor comparisons that lead to new calibrations, VIIRS and MODIS would not return the same values of AOD (Sawyer et al., 2020). If we view the globe with multiple GEO sensors that never overlap, the three sensors will diverge, if not immediately on orbit, eventually as their sensors degrade and are re-calibrated over time. This re-calibration is complicated depending on wavelength, angle, and solar exposure but can require corrections of 20-30% (https://digitalcommons.usu.edu/calcon/CALCON2015/All2015Content/39/). The fact that we have polar orbiting sensors flying through the GEO disks on a daily basis gives us the ability to bring the entire observing system into a degree of conformity.

Levy, R. C., Mattoo, S., Sawyer, V., Shi, Y., Colarco, P. R., Lyapustin, A. I., Wang, Y., and Remer, L. A.: Exploring systematic offsets between aerosol products from the two MODIS sensors, Atmos. Meas. Tech., 11, 4073–4092, https://doi.org/10.5194/amt-11-4073-2018, 2018.

Sawyer, V.; Levy, R.C.; Mattoo, S.; Cureton, G.; Shi, Y.; Remer, L.A. Continuing the MODIS Dark Target Aerosol Time Series with VIIRS. Remote Sens. 2020, 12, 308. https://doi.org/10.3390/rs12020308

Page 3, Line 13-14: If Dark Target is applied to different sensors I assume it needs different parameterizations.

Yes. Because of the different wavelengths explained in Table 2, each sensor requires its own Look Up Table. Because of the different spatial resolutions, the internal process of organizing the sensor observations is different. However, we believe that the reviewer is asking us about the land surface parameterization that links surface reflectance in the visible wavelengths to surface reflectance in the 2 µm spectral region. These parameterizations are being evaluated for the new sensors (Kim et al., 2024), but the retrievals used in this study rely on the same parameterization adapted for MODIS. This assumes the relationship between blue, red, and 2 µm surface reflectance holds no matter which specific blue, red, or 2 µm wavelength we are using. This is correct for the first order. We find in Kim et al. (2024) that errors introduced by the surface parameterization are larger due to differences in GEO geometry than in differences in specific wavelengths.

Kim, M., Levy, R. C., Remer, L. A., Mattoo, S., and Gupta, P.: Parameterizing spectral surface reflectance relationships for the Dark Target aerosol algorithm applied to a geostationary imager, Atmos. Meas. Tech., 17, 1913–1939, https://doi.org/10.5194/amt-17-1913-2024, 2024.

Results and Discussions

Figure 1 seems to remark global coverage by combining the six satellite sensors. However, it does not remark the main goal of improving temporal coverage with high spatial resolution.

We believe we have two goals. One is filling in holes caused by clouds with at least one observation per day. This is spatial coverage. The second goal is to resolve the diurnal signature. Agreed that Figure 1 only supports the first goal. We have now explicitly stated these two goals in the Introduction.

Figure 2: Why is this Figure relevant?

Figure 2 provides an estimate of level 2 data pixel size for each sensor used in the study. The figure is discussed in section 3.3 under Caveat # 1. The pixel size is important information for data users to consider while applying data from multiple sensors. Therefore, the figure provides transparency about datasets and enables users to assess whether this data is suitable for certain applications. We also use this information in our data processing to avoid any data gaps arising due to the large pixel size at the edge of the swath (as discussed in caveat #1).

Figure 3: This Figure needs further discussion, particularly for AOD below 0.1.

We have added text to clarify the purpose and description of the figure. The primary purpose of this figure is to demonstrate the spatial coverage. The quantitative analysis is provided in comparison charts.

Figure 5: Further discussion is need about the fact that larger deviations are observed for LOE sensors.

The differences among LEO sensor are well documented in previous studies ((Levy et al. 2018, Lyapustin et al, 2023; Schutgens et al, 2020, Sogacheva et al., 2020) and consistent with current study.

We further clarify the text to explain the reasoning and now write,

"The LEO sensors' deviation from the value of 1 is primarily due to the relatively poor spatial-temporal sampling of these sensors within the specific GEO-region."

Figure 7: It is straightforward that the merged product has more measurements. Also, Europe/Africa region is less represented, but there are also geostationary observations for these regions.

Europe and Africa are covered by an older generation of geostationary satellite sensors (SERVIR), which does not have sufficient spectral bands to apply the DT algorithm. The newly launched GEO sensor FCI on Meteosat third generation is similar to ABIs and, therefore, will be able to fill those gaps in future versions of this dataset. We mention that these other sensors could fill in additional holes in the future at the end of the conclusion section.

Figure 10: Is there a better agreement with AERONET for AOD merged products? It is not clear. Indeed, the 1:1 results look quite similar.

We don't expect a better agreement with AERONET for the merged product. The agreement is fairly similar to individual sensors and closer to GEO than LEO. Overall, LEO sensors are more mature and have better comparisons compared to GEO sensors. We further clarify this in the text.

Conclusions:

Page 11, Line 26: High quality aerosol observations? To my knowledge the only reliable product is AOD.

We believe that it is understood that the past LEO sensors that we mention are limited in aerosol characterization so that AOD is the most reliable of the retrieved products and spectral AOD leading to a size parameter is also reliable over ocean. We don't see the necessity of spelling that out here.

Page 11, Line 32: Does this study implement Dark Target? The algorithm and its application are well known in the scientific community.

Yes, the study uses data from the implementation of same version of DT algorithm on all six sensors together.

Page 12, Line 14-15: There is no demonstration of the merged product for tracking aerosol transport / variability in a single place.

Agreed, we don't provide an example here. The focus of this study is to describe new datasets; therefore, it is beyond the scope of the study to demonstrate all potential applications. We hope that the user community will use the new datasets to understand transport patterns and apply them to other applications. In fact, our team is analyzing these datasets to address various science and application questions, which will be published in future articles.

Page 12, Line 18-19: From the result presented in the manuscript I have not seen an unexpected strong diurnal signature from the global composite of AERONET AOD during northern Spring

We refer the reviewer to Figure 12. The AOD varies from about 0.05 in the morning to 0.3 in the afternoon. We further revised the text to clarify it.

---

## Author Comment (AC2)

**RC1**: 'Comment on amt-2023-259', Anonymous Referee #1

**General Comment:**

The authors apply the dark target aerosol optical depth algorithm to six satellite instruments (3 geo, 3 leo). They produce a quarter degree gridded product with statistics for each instrument and an ensemble average. They use the gridded product for intercomparison and validation against AERONET. The manuscript is a significant contribution that fits well within the scope of AMT. I have only minor comments that focus on methodological clarity and minor editorial comments. The resultant data product will likely be extremely valuable to multiple air pollution disciplines.

**Response:**

We thank the reviewers for insightful comments that have improved the manuscript significantly. We have provided response to each comment below.

Specific Comments: (original comments are in black and responses are in blue)

pg1, 22-23, I suggest moving the correlation before the percent within EE because it makes it could be read that the correlation is related to that subset.

modified as suggested.

pg1, 44: grid[d]ed

checked and corrected throughout the manuscript.

pg2, 16: my copy shows a strikeout that should be addressed.

identified and fixed

pg2, 16: SNPP and Aqua seem close in time, but Terra seems like a meaningfully different overpass time.

modified as follow:

"Having multiple polar-orbiting views of the same scene might increase data product availability, but not much if the two instruments pass close in time, such as do Aqua, and S-NPP in North America."

pg2, 20-21: As written, this excludes the main reasons for missing pixels and then concludes nearly complete... The no clouds *and otherwise retrievable* seems weird.

text revised to clarify it.

"In fact, because a scene is rarely continually cloudy from sunrise to sunset, we see that geostationary sampling can find at least one cloud-free opportunity to make an aerosol retrieval on any day, This increases the probability of at least one aerosol retrieval sometime during the day to nearly 100%  (Remer et al., 2012)."

pg3, 30: (ATBD, 2023[)]

edited

pg4, 18: Can you be more specific about "after some time"? Are we talking about Phase F or something earlier?

text revised to clarify it.

"At first they produced very similar results (Remer et al. 2006), but after the implementation of Collection 5 calibration and data processing that began in September 2006 the DT aerosol results from the two sensors began to diverge (Levy et al. 2018)."

pg4, 25: Section 3 really only addresses LUT updates. Are algorithm adjustments always LUT updates? Or are there any more substantial updated?

This is covered in section 3.0. Table 2 provide specific differences in each sensor. We had added text to point out additional aspects more specifically such as pixel size, cloud masking, coverage, etc.

pg5, 14: It would be good for Table 2 or the text to explicitly mention overpass times.

Added information on equatorial overpass time in Table 2

pg5, 34-35: Are any of the AERNET not collocated with leo orbits?

both LEO and GEO sensors are collocated with AERONET stations in their respective coverage area.

pg6, 18: viewing "angle" will vary by product.

Yes, view angle varies for each sensor. For each GEO sensor, the viewing geometry is fixed, while for LEO it changes for every orbit and only repeat once every 16 days.

pg6, 34: Are you saying finer pixel measurements at nadir are aggregated so that the pixel size range is smaller? Is that what the jumps are in Figure 2?

The text is revised, and following reference is added to further explain VIIRS pixel aggregation:

Elvidge, C.D.; Zhizhin, M.; Hsu, F.-C.; Baugh, K.E. VIIRS Nightfire: Satellite Pyrometry at Night. *Remote Sens.* **2013**, *5*, 4423-4449. https://doi.org/10.3390/rs5094423

The figure below (from Elvidge et al., 2013) and caption explains VIIRS different aggregation regimes.

[Figure]

Horizontal sample interval chart shows how the growth of VIIRS M-band ground field of view from nadir to edge of scan is constrained by switching the number of pixels that are aggregated [12]. In aggregation zone 1, from nadir to 31.72 degrees, the signal from three pixels are averaged. In aggregation zone two, the signal from two pixels are averaged. In aggregation zone three, signal from a single pixel is recorded.

pg6, 38: "box gridding" is not a term I am used to. Is this referring to binning pixels based on their centroids being within a quarter degree cell (nearest neighbor based on centroids)?

The box griding is discussed in our previous publication (Gupta et al., 2020) where averaging method is explained in great detail. In simple words, "box gridding" here refers to everybody's simple concept of an average of all the pixels with center lat-lon that fall within the 0.25x0.25 degree latitude and longitude box for each grid. Because some MODIS pixels are 40 km apart, we end up with grid boxes having no MODIS

centers within the box. This is compounded by bow tie effects that can place two 40 km pixel centers closer together and another two 40 km pixels further apart than the nominal 40 km. Modified text to clarify this.

pg6, 39: "spatial filling method" as described sounds like "averaging pixels whose footprint overlaps a grid cell".

Some sensor's pixels at the edge of the swath can cover multiple quarter degree grids of measurements due to their large size, therefore, simple averaging can create artificial gaps in gridded data due to 0.25x0.25 restriction. Therefore, to avoid these gaps, a simple gap filling method based on viewing angle and pixel size is adopted so that grids represent actual measurement footprints without any gaps in data. Gupta et al., 2020 describe this in more details with examples.

pg7, line 21: Visible discontinuity at the scale displayed seems like an unreasonable metric. We'd expect the discontinuity to be larger for a single scene when zoomed in.

Yes, agreed. Therefore, we specify it as qualitative and later we demonstrate quantitative differences among sensors.

pg7, 37-38: This seems like a weird choice. I agree that it likely doesn't change the conclusions, but a 1 in 30 sample seems like an unnecessary simplification.

One day per month is selected to ensure sampling represents all the seasons as well as data presented are manageable. As we increase to number of days, volume of data become challenging to put on scatter plot. But we agree that the conclusion will not change. We further clarify this in the text.

pg8, 4: The g17 also looks at the arid west where aod comparisons have revealed higher uncertainty. I think it is important to note that it isn't just US vs Asia, but within countries as well.

Agreed. We added text to further clarify this.

Meanwhile, ABI-G16 covers the generally wetter and darker eastern North and South America, while ABI-G17 covers mostly ocean and the dryer and brighter western North America. The significant differences in surface type will affect the accuracy of the retrievals.